# A Modified Wet Transfer Method for Eliminating Interfacial Impurities in Graphene

**DOI:** 10.3390/nano13091494

**Published:** 2023-04-27

**Authors:** Dong Jin Jang, Mohd Musaib Haidari, Jin Hong Kim, Jin-Yong Ko, Yoonsik Yi, Jin Sik Choi

**Affiliations:** 1Department of Physics, Konkuk University, Seoul 05029, Republic of Korea; 2Superintelligent Creative Research Laboratory, Electronics and Telecommunication Research Institute (ETRI), Daejeon 34129, Republic of Korea

**Keywords:** graphene transfer, FeCl_3_, interfacial impurity, clean interface, Janus interface

## Abstract

Graphene has immense potential as a material for electronic devices owing to its unique electrical properties. However, large-area graphene produced by chemical vapor deposition (CVD) must be transferred from the as-grown copper substrate to an arbitrary substrate for device fabrication. The conventional wet transfer technique, which uses FeCl_3_ as a Cu etchant, leaves microscale impurities from the substrate, and the etchant adheres to graphene, thereby degrading its electrical performance. To address this limitation, this study introduces a modified transfer process that utilizes a temporary UV-treated SiO_2_ substrate to adsorb impurities from graphene before transferring it onto the final substrate. Optical microscopy and Raman mapping confirmed the adhesion of impurities to the temporary substrate, leading to a clean graphene/substrate interface. The retransferred graphene shows a reduction in electron–hole asymmetry and sheet resistance compared to conventionally transferred graphene, as confirmed by the transmission line model (TLM) and Hall effect measurements (HEMs). These results indicate that only the substrate effects remain in action in the retransferred graphene, and most of the effects of the impurities are eliminated. Overall, the modified transfer process is a promising method for obtaining high-quality graphene suitable for industrial-scale utilization in electronic devices.

## 1. Introduction

The integration of graphene into optoelectronic applications requires a large, homogeneous graphene sheet that is typically produced by chemical vapor deposition (CVD) on metal catalysts such as Cu or Ni, producing diverse outcomes from monolayer to multilayer graphene [1,2,3]. For graphene to be used in electronic devices, it needs to be transferred onto substrates. The key process in conventional wet transfer involves coating graphene with a supporting polymer layer and etching the metal catalyst. Various etchants have been used to etch transition-metal catalysts [4,5], including iron chloride (FeCl_3_), which is widely used as a metallic etchant to etch various metals such as copper and nickel. However, these processes introduce two types of impurities into the graphene layer: polymer residues on the graphene surface and interfacial impurities [6,7,8,9,10]. Eliminating interfacial impurities is the most challenging task because they are encapsulated between the impenetrable graphene layer and the substrate [11].

The use of FeCl_3_ as a metal etchant can lead to the formation of complex nanoparticles at the interface between graphene and the substrate. Kraus et al. reported that FeCl_3_ etching of copper resulted in cubic Fe_0.942_O nanoparticles under graphene, indicating the presence of uncontrolled phases of iron oxides such as FeO, Fe_2_O_3_, Fe_3_O_4_, etc. These iron oxides can exhibit n- or p-type behaviors depending on the phase species [12,13,14]. Several alternative methods have been proposed to address this problem, including electrochemical delamination [15], mechanical delamination [16], and RCA cleaning [17]. However, each of these methods has limitations, such as requiring careful control of the voltage and electrolyte concentration, introducing the potential for mechanical damage to the graphene, necessitating the usage of hazardous solutions, and requiring time-consuming additional processing [18].

In this study, we introduce a modified wet transfer method that uses a temporary SiO_2_ substrate to achieve high efficiency in eliminating impurities from the graphene/substrate interlayer. In this technique, the graphene layer is first transferred onto a UV-irradiated hydrophilic substrate, allowing water to intercalate at the interface between the graphene and the substrate. In this process, interfacial impurities are retained on the temporary substrate, and the cleaned graphene can be retransferred to a target substrate without physical damage. Using optical microscopy and Raman spectroscopy, we confirmed that all interfacial impurities in the retransferred graphene were eliminated. Furthermore, we demonstrated significant improvements in electrical measurements using the transmission line model (TLM) and Hall effect measurements (HEMs), such as a reduction in sheet resistance and electron–hole asymmetry in the retransferred graphene. We suggest that this modified wet transfer method can be implemented as an efficient method for obtaining a clean interface during the transfer of all thin films and 2D materials, as well as for devices that are sensitive to interfacial properties [19,20,21,22].

## 2. Experiments

### 2.1. Chemical Vapor Deposition of Graphene

Graphene was grown on commercial copper foils (99.95% purity, 35 μm thickness, Graphene Platform^TM^) in a tube-type CVD. The Cu foil was placed inside the CVD chamber and annealed for 1 h with 200 sccm of flowing argon gas, raising the chamber temperature to 980 °C. Subsequently, 80 sccm of hydrogen gas was introduced, and the sample was annealed at the remaining temperature for 1 h. Following annealing, 1.5 sccm of methane gas was introduced to grow graphene for 2 h, resulting in continuous single-layer graphene. Following the growth process, the graphene was rapidly cooled by moving the furnace to the cooling zone.

### 2.2. Substrate Treatment

Thermally oxidized SiO_2_ (300 nm)/Si substrates were prepared using various surface treatments, including blowing N_2_ gas, sonication in deionized (DI) water (40 min), sonication in acetone (20 min) and isopropyl alcohol (20 min), UV treatment (60 s), and annealing at 120 °C for 30 min in high vacuum, to optimize the substrate cleaning conditions for electrical property measurements [23].

### 2.3. Raman Spectroscopy

Raman spectroscopy was performed using a confocal spectrometer (UHTS 300, WITec GmbH, Ulm, Germany) at the CORE Facility Center for Quantum Characterization/Analysis of Two-Dimensional Materials and Heterostructures. The measurements were recorded with a 532 nm laser with 5 mW power and a 100× objective, resulting in a laser spot size of approximately 2 µm.

### 2.4. Device Fabrication

Using the conventional wet transfer method and our modified retransfer method, graphene was transferred onto SiO_2_/Si substrates with pre-deposited Cr/Au (5 nm/30 nm) electrodes. The electrode and graphene channels were defined using laser lithography (4PICO BV^TM^, PicoMaster 200, Sint-Oedenrode, Netherlands) with KL5305 (KemLab^TM^, Woburn, MA, USA) as the photoresist and PMMA 950 A6 (Microchem^TM^, Microchem, Newton, MA, USA) as the buffer layer.

## 3. Results and Discussions

Figure 1a illustrates the modified wet transfer process for graphene, which involves an additional retransfer step through spontaneous delamination. The process began with a conventional method for graphene growth and transfer, in which a graphene-grown Cu foil synthesized by tube-type CVD was prepared. To transfer graphene, poly (methyl methacrylate) (PMMA) was spin-coated as a supporting layer, and backside etching using O_2_ plasma was performed. The Cu foil was etched by floating it on an iron chloride (FeCl_3_) solution followed by rinsing it with DI water. The general wet transfer process typically involves transferring the PMMA/graphene structure to the final target substrate and removing the PMMA after drying. However, we modified the process by transferring the PMMA/graphene structure onto a temporary SiO_2_ substrate that had been treated with UV light. Several treatments were tested to determine the highest hydrophilicity for the temporary substrate, among which UV treatment was the most efficient (see Appendix A). This surface modification from hydrophobic to hydrophilic enables the spontaneous separation of the PMMA/graphene structure from the substrate via water intercalation, even after the interface between graphene and the substrate has dried. This spontaneous delamination marks the beginning of the retransfer process and is enabled by extraordinarily long-wavelength capillary fluctuations when water is confined between the adjoining hydrophobic and hydrophilic surfaces. Xueyun et al. explained dynamic fluctuations in confined water at the Janus interface [24].

Graphene is a well-known hydrophobic material, and the UV-treated SiO_2_ surface maintains its hydrophilicity during the initial transfer onto a temporary substrate (see Appendix A). Following the drying process during the initial transfer (30 min < t < 2 h), the PMMA/graphene/temporary substrate was dipped into DI water. The PMMA/graphene was then separated by spontaneous water intercalation owing to the Janus interfacial effect. This was not the first transfer because multiple transfers without drying between newly supplied DI water were conducted in the previous rinsing process. However, using a temporary substrate with a hydrophobic surface causes graphene to adhere to the substrate when it dries. The most interesting aspect of this process is that when Janus interfacial water intercalation is used, PMMA/graphene, which appears to adhere to the substrate when dried, is delaminated, leaving impurities on the hydrophilic surface. This contribution is due to the hydrogen bond mechanism, which is hydroxyl–hydroxyl bonding between the groups on the metal oxide surface and UV-treated SiO_2_ surface [25], which is ~5 times stronger than the van der Waals force between hydrophobic graphene and hydrophilic metal oxide surface. In Figure 1a, we designate black, red, and blue checkmarks, and their corresponding optical microscopy images are shown in Figure 1b–d. Low-magnification optical microscopy images revealed the topography of the coated PMMA, including the positions and shapes of the interfacial impurities. The high-magnification optical images in the dotted area show impurity transfer from the graphene surface to the temporary substrate. These effects were confirmed by AFM imaging as illustrated in Appendix A. As can be seen in Figure 1a–d, material integrity was maintained, and no additional wrinkles were generated when additional transfer processes were carried out. Finally, interfacially cleaned graphene was obtained using a retransfer method. Table 1 shows the advantages and disadvantages of previously reported transfer methods.

To compare the quality of the two different graphene samples obtained using the conventional transfer and retransfer methods, we conducted optical microscopy. As shown in Figure 2a,e, the optical microscopy images confirm a noticeable improvement in cleanliness (see Appendix A for a comparison of the whole-size optical microscopy images). This improvement was also observed in samples at a cm scale. When transferring graphene onto a target substrate, various factors must be considered to improve the quality of the resulting graphene. These factors can be categorized into visible metal impurities and PMMA residues, as well as invisible strain and dopant effects at the graphene–substrate interface and graphene surface [26,27]. Although both factors are important, interfacial impurities are particularly challenging to address because they become inaccessible after the PMMA/graphene adheres to the substrate through the drying process. Using the retransfer method, we eliminated the interfacial impurities by transferring them to a temporary substrate during water-induced delamination. Optical microscopy observations confirmed that visible impurities were effectively eliminated; however, improvements in the invisible strain and doping effects remain to be identified.

To check for the invisible quality, we performed Raman analysis of the transferred and retransferred graphene. Raman analysis was conducted by mapping an area of 50 × 50 μm^2^ consisting of Raman spectra from 50 × 50 points. As shown in Figure 2a,e, the Raman mapping results are expressed as the intensity ratios of the D and G peaks (I_D_/I_G_) to the 2D and G peaks (I_2D_/I_G_). Compared with the optical microscopy results, the local impurity points and line-shaped graphene folds have high intensities in the I_D_/I_G_ mapping but low intensities in the I_2D_/I_G_ mapping. Generally, measuring Raman spectra of graphene on a metal substrate is the most challenging task because the scattering of the metal surface hinders the identification of the distinct peak information of graphene. If graphene is transferred onto a partially metal-containing substrate, that is, onto metal nanoparticles, the background signal increases and shades the graphene spectrum [28,29]. To compensate for this background effect, we used the intensity ratios I_D_/I_G_ and I_2D_/I_G_, instead of I_D_ and I_2D_ alone. In general, I_D_ (~1350 cm^−1^) indicates a defective non-sp^2^ domain, and some iron oxide phases exhibit a distinct peak near ~1320 cm^−1^ [30]. Therefore, I_D_/I_G_ can be used as an index that shows the position of the residual impurities and their structural effects on graphene. I_2D_/I_G_ is used as an index not only for confirming that the graphene is single-layer, but also for determining the amount of p-doping owing to the increase in the G-peak and decrease in the 2D-peak on p-doped graphene [31].

We compared the positions, distributions, and electrical effects of the impurities between the two methods. Three points were selected for each Raman mapping result. In the transfer result (Figure 2a), the impurities were clearly distinguishable; therefore, we chose two positions where impurities existed (i and ii) and one flat region (iii). Because there were no distinct impurities in the retransfer results (Figure 2e), we chose two line-shaped graphene folds (iv and v) and one flat region (vi). Figure 2b,f show the line profiles obtained at the same positions from the D-, G-, and 2D-mapping results. Because all Raman intensities are normalized by the Si-peak (I_Si_) to 1, we could observe changes in both the Raman intensity and intensity ratios. Interestingly, in contrast to the D-peak intensity, which increased due to local defects, the G- and 2D-peak intensities also increased as a result of the impurities. This is explained in Figure 2c, where the impurity-existence positions exhibited high background Raman spectral features throughout the Raman shift range. This feature is clearly distinguished from the simple structural disorder of the folds shown in Figure 2f,g. Based on the results, these interfacial impurities are known to have metallic features, as reported in previous papers [32]. Furthermore, through statistical analysis of the distribution and average value changes shown in Figure 2d,h, we found that the retransferred graphene is expected to have improved electrical properties compared to the other methods (normalized properties).

To quantitatively analyze the improvement in strain and doping, we used a vector decomposition method based on the quasi-linear change in phonon energy with strain and doping [33]. In this vector space, each given point in ($\omega_{G}$, $\omega_{2D}$) space can be translated into an $\vec{OX}$ vector, which is shifted from an intrinsic point O (1582 cm^−1^, 2676.7 cm^−1^) [34]. The $\vec{OX}$ vector can then be decomposed into a tensile strain vector $\vec{OT}$ using $\vec{e_{T}}$ basis and a hole-doping vector $\vec{OH}$ using an $\vec{e_{H}}$ basis. As shown in Figure 3a, the strain of the retransferred graphene (red crosses) improved from the tensile strain of the transferred graphene (black cross marks) to compressive strain. In most cases, exfoliated and transferred graphene exhibits compressive strain states [35,36], but the lifting pressure by impurities could be the origin of this tensile strain. In contrast to this significant change in the strain, doping did not result in significant differences.

To determine the strain and doping, we used the Jacobian transformation (Equation (1)) [37].
(1)$$\vec{OX}=\frac{x_{G}h_{2D}-x_{2D}h_{G}}{t_{G}h_{2D}-t_{2D}h_{G}}\vec{e_{T}}+\frac{x_{2D}t_{G}-x_{G}t_{2D}}{t_{G}h_{2D}-t_{2D}h_{G}}\vec{e_{H}},$$
where $x_{G}$ and $x_{2D}$ are components of an arbitrarily measured $\vec{OX}$ vector, a shift from origin in G-peak and 2D-peak energy in ($\omega_{G}$, $\omega_{2D}$) space; $t_{G}$ and $t_{2D}$ are the shifts of the G- and 2D-mode energy as a function of tensile strain; and $h_{G}$ and $h_{2D}$ are the shifts of the G- and 2D-mode energy as a function of hole doping, respectively.

Using Equation (1), we reconstructed Raman mapping images of the disentangled strain and doping of the transferred and retransferred graphene (Figure 3b). Interestingly, we observed tensile strain in a larger area near the impurity-containing points (i and ii), as well as in the flat region (iii). This can be interpreted as the presence of metallic particles causing each carbon atom to move farther apart, resulting in tensile strain in the graphene. This effect is similar to that observed in the formation of graphene bubbles under positive pressure [38,39]. In contrast, the folded regions (iv and v) in the retransferred graphene showed a compressive nature resulting from the CVD growth and folding process [40], whereas the flat region (vi) did not show a notable change in compressive strain compared to other areas. Our results suggest that impurity-containing graphene can be improved to the level of cleanly transferred graphene by adding a retransfer process. This improvement was verified through statistical analyses. In the disentangled Raman mapping of doping, we observed p-doping at certain impurity spots where metallic particles were located in the impurity-containing regions, which is consistent with the fact that several phases of iron oxide are p-type semiconductors that accept electrons [13]. Additional impurity spots that are not discernible in the I_D_/I_G_ analysis, and are visible in the tensile strain analysis, exhibit n-type doping characteristics, indicating electron donation. This may be attributed to the n-type behavior of certain iron oxide phases [12]. Overall, the average doping distributions of the transferred and retransferred graphene samples did not exhibit significant differences. However, it should be noted that impurities have numerous undefined phases, which makes their effects on doping unclear.

The electrical properties of graphene were investigated to confirm the effect of the retransfer process using transmission line model (TLM) devices with a width of 50 μm and a total length of 800 μm. Larger channel dimensions were chosen to avoid sample-to-sample variations due to the impurity size, and the distances between the impurities were at the micro and sub-micro scales. As shown in Figure 4a, the graphene was transferred to Cr/Au (5 nm/30 nm) metal electrodes patterned onto a Si/SiO_2_ substrate. Laser lithography was used with a photoresist (PR) and PMMA heterostructure to pattern the electrodes and etch graphene to define the channel dimensions. In general, e-beam lithography is preferred for graphene device fabrication because of its many advantages in fabricating submicropatterns and effective elimination of the PMMA layer. Although laser lithography is faster for fabricating large-scale patterns, an effective cleaning method for PR residues on graphene has not yet been developed. Therefore, we spin-coated PMMA onto the graphene surface and sequentially deposited PR. After developing the desired pattern on the PR, the PMMA and underlying graphene layers were etched using O_2_ plasma. Subsequently, the remaining PR/PMMA layers were removed using acetone/IPA solvent.

The I_D_-V_G_ curves for the transferred and retransferred graphene devices and their respective channel lengths are presented in Figure 4b,c, respectively. A gate voltage sweep was performed between −40 V and +80 V with a drain-source bias of 10 mV. As expected from the OM and Raman analyses, the retransferred graphene showed stable electrical properties, whereas the transferred graphene exhibited very unstable electron–hole asymmetry distributions. This can be explained by the neutral-scattering effect [41] from the various n- and p-type dopant distributions. Hence, lower current levels and uneven Dirac voltages arise from the uneven distribution of impurities consisting of mixed phases of iron oxides, which provide n- and p-type dopants. This explanation is in agreement with the analysis in Figure 3, which implies that visible impurities have strong p-doping and n-doping effects in the flat region. These complex effects are reflected in the ratio of electron and hole mobilities (μ_e_/μ_h_) of transferred graphene FET, which averages 0.76 from 10 devices. In Figure 4c, we obtained the I_D_-V_G_ curve for the retransferred graphene, which showed relatively stable Dirac voltages and drain current levels. This suggests that the interfacial doping effects were significantly reduced, and only the SiO_2_ substrate effects remained active [42]. Generally, graphene transferred onto SiO_2_ tends to undergo p-doping. For the retransferred graphene, the ratio between electron and hole mobilities (μ_e_/μ_h_) averaged 0.85.

To examine the changes in the conductance and scattering effect of the TLM devices, I_D_-V_D_ measurements were performed with V_D_ = 10 mV. The resulting sheet resistance (R_S_) is shown as a function of channel length in Figure 4d. The equation R_Total_ = 2R_c_ + R_s_ L/W was used to fit the data, where R_C_ is the contact resistance, L is the length, and W is the channel width. We also examined these effects using Hall effect measurements with the van der Pauw method to investigate the electrical properties of the graphene sheet, such as the R_S_, carrier concentration, and Hall mobility [43]. (See Appendix A.) As a result, the RS of the retransferred graphene was reduced by 23% compared with that of the transferred graphene. Additionally, Dirac voltage-dependent hole mobilities are plotted in Figure 4e, which were extracted using the equation for graphene FET, μ = (dI_D_/dV_G_) (L_ch_/W_ch_V_D_C_ox_) [44], where SiO_2_ capacitance is C_ox_ = 11.51 × 10^−9^ F/cm^2^. The black and red circles indicate the spread areas of the hole mobilities versus the Dirac voltages for the transferred and retransferred devices, respectively. Comparing the circles, the retransferred sample exhibits a larger improvement than the transferred graphene. We found that the impurities in the graphene–substrate interface consist of p- and n-type dopants with random distributions, resulting in wide spreads in the Dirac voltage and carrier mobilities of the transferred graphene.

## 4. Conclusions

Our study demonstrated the effectiveness of a modified wet transfer technique for eliminating interfacial impurities in CVD-grown graphene. Using a temporary SiO_2_ substrate and UV treatment, our retransfer process enabled the removal of metallic impurities and reduced the interfacial doping effects in graphene. As a result, the retransferred graphene showed significantly improved electrical properties, including reduced sheet resistance and stable electron–hole asymmetry distributions. These findings provide valuable insights for the development of high-performance graphene-based electronic devices, highlighting the importance of interfacial purity in achieving optimal device performance.

## Figures and Tables

**Figure 1 nanomaterials-13-01494-f001:**
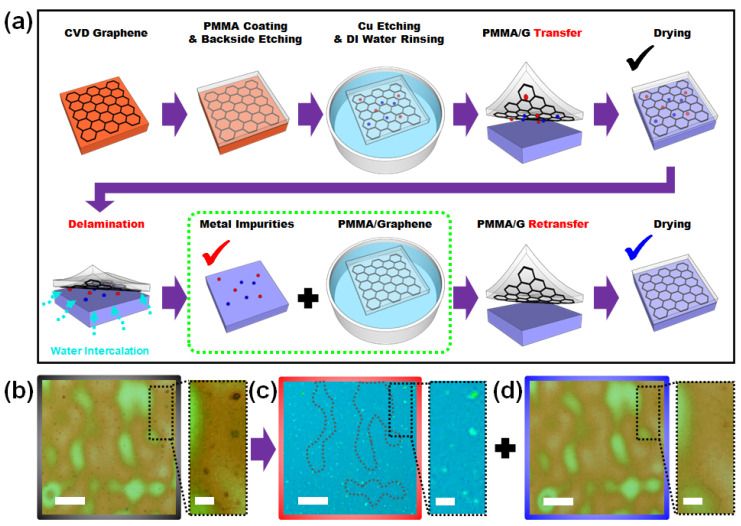
Retransfer process of a CVD-grown graphene. (**a**) Schematic diagram of transfer and retransfer processes. The transfer process depicted in the first row exhibits the general transfer process of CVD-grown graphene, and the second row exhibits our retransfer process. (**b**–**d**) Optical microscopy images of the PMMA/graphene transferred onto a temporary SiO_2_ substrate (**b**), the temporary SiO_2_ substrate after delamination by water intercalation (**c**), and the retransferred PMMA/graphene onto a final target SiO_2_ substrate. The black, red, and blue checkmarks in (**a**) correspond to the resultant optical microscopy images shown in (**b**), (**c**), and (**d**), respectively. The scale bar for (**b**–**d**) is 20 μm, while the magnified optical images of the dotted inset area have a scale bar of 5 μm.

**Figure 2 nanomaterials-13-01494-f002:**
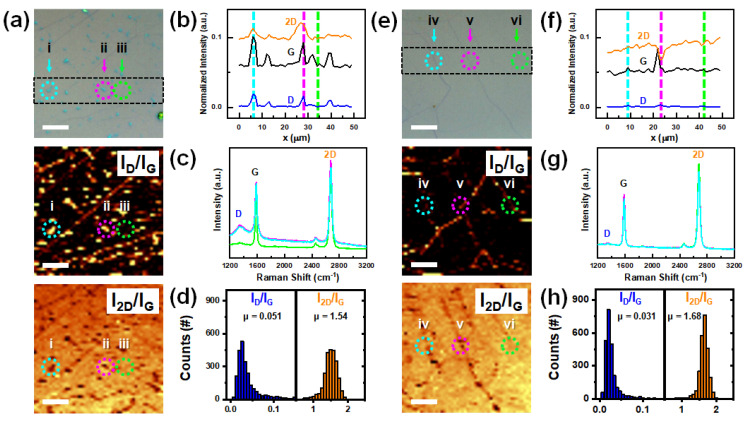
Raman analysis of graphenes obtained by transfer and retransfer methods. Raman mapping was performed in an area of 50 × 50 μm^2^, with 50 × 50 resolution using a 532 nm wavelength laser. (**a**) Optical microscopy image, I_D_/I_G_ and I_2D_/I_G_ Raman intensity ratio mapping results of the graphene obtained through the conventional wet transfer method. (**b**) Line profiles of Raman intensity extracted from normalized D-, G-, and 2D-intensity mappings in dotted rectangle designated in optical microscopy image in (**a**). All intensities in the Raman spectra analysis are normalized with Si-peak to 1. (**c**) Raman spectra extracted from each position of (i), (ii), and (iii) in the dotted circle indicated in (**a**). (**d**) I_D_/I_G_ and I_2D_/I_G_ ratio distributions with their average values (μ). (**e**–**h**) Same set with (**a**–**d**) for the graphene sample obtained through the retransfer method. (iv) and (v) are defective regions, and (vi) represents non-defective regions. Scale bar is 10 μm.

**Figure 3 nanomaterials-13-01494-f003:**
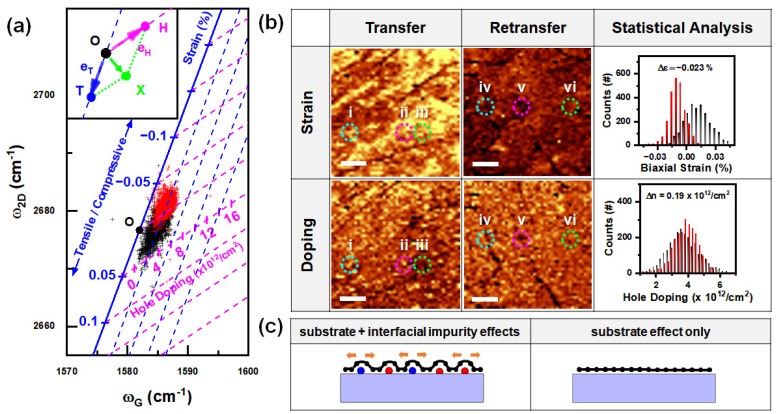
Doping and strain comparisons between transferred and retransferred graphene. Vector decomposition analysis was used to construct doping and strain independently [33]. (**a**) Correlation between the energy shifts of the G- and the 2D-modes. The 2500 spectra obtained at the mapping results in Figure 2 were analyzed using a reference position O, which was introduced as the intrinsic energies of the G- and 2D-modes from [34]. The inset shows the designation of the vector analysis. The $\vec{OX}$ vector represents the energy shifts of the G- and 2D-modes from the origin of each spectrum. The $\vec{OH}$ vector represents decomposed hole-doping vector of $\vec{OX}$, using $\vec{e_{H}}$ as a unit vector. The $\vec{OT}$ vector represents the decomposed tensile strain vector of $\vec{OX}$, using $\vec{e_{T}}$ as a unit vector. The G- and 2D-mode energies of the transferred graphene and the retransferred graphene are designated with black- and red-cross symbols, respectively. The dashed doping and strain lines are denoted in blue and pink, respectively [37]. (**b**) Comparisons of the strain and doping analysis for the transferred graphene (black) and the retransferred graphene. (**c**) Illustrations comparing the interface states and their effects between the transferred graphene and the retransferred graphenes.

**Figure 4 nanomaterials-13-01494-f004:**
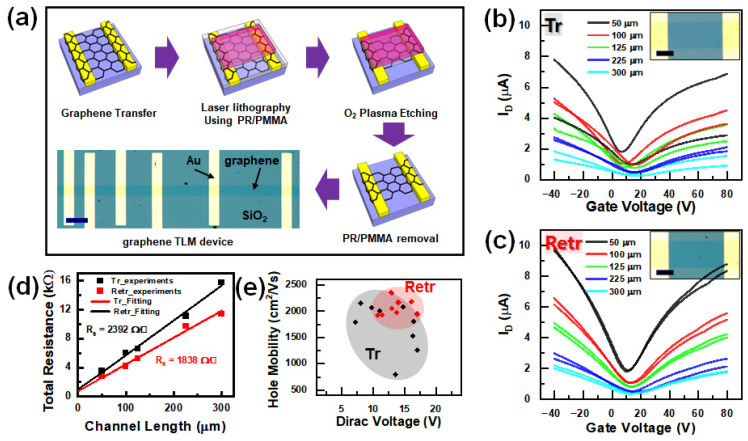
Transport characteristics of transferred and retransferred graphene. (**a**) Schematic of device fabrication process and OM image of resulting device. Scale bar is 100 μm. (**b**,**c**) FET characteristics of (**b**) conventional wet transferred (**c**) retransferred graphene. The drain voltage is 10 mV for all measurements. (Inset) magnified OM image of each graphene device scale bar is 20 μm. (**d**) TLM characterization plots of differently transferred graphene. (**e**) Dirac voltage and hole mobility correlation. The distributed area is denoted around each datum with a corresponding-colored ellipse.

**Table 1 nanomaterials-13-01494-t001:** Comparison of different graphene transfer methods.

Transfer Method	Advantages	Disadvantages	Refs.
Conventional wet transfer	Easy and scalable	Presence of metallic impurities	[8]
Electrochemical delamination	Free of metallic impurities; metal catalyst reusable	Additional instruments required; damage caused by bubbles	[8,15]
Mechanical delamination	Free of metallic impurities; metal catalyst reusable	Time-consuming; hard to scale up; may cause mechanical damage	[16]
RCA clean	Free of metallic impurities	Use of hazardous solution; time-consuming; generation of metal waste	[17]
Modified wet transfer (This work)	Free of metallic impurities; easy and scalable	Generation of metal waste	

## Data Availability

Data can be available upon request from the authors.

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
