# Peer review of "A Modified Wet Transfer Method for Eliminating Interfacial Impurities in Graphene"

_nanomaterials, 2023, doi:10.3390/nano13091494_

Round 1

Reviewer 1 Report

The authors proposed a modified wet transfer method for graphene using a UV-treated SiO2 substrate to absorb the impurities. The retransferred graphene showed good quality. The results are interesting. I would like to recommend the publication of this paper.

Author Response

Thank you for your positive feedback and recommendation for publication. We appreciate your interest in our work.

Reviewer 2 Report

In this study, the authors introduced an improved transfer process that utilizes a temporary SiO2 substrate treated with ultraviolet light to adsorb impurities from graphene and then transfer them back to the final substrate. Optical microscopy and Raman mapping confirm the adhesion of impurities onto the temporary substrate, leading to a clean graphene/substrate interface. These results indicate that only substrate effects remain in action in retransferred graphene, and most of the effects from impurities are eliminated. Overall, the modified transfer process is a promising method for achieving high-quality graphene suitable for industrial-scale utilization in electronic devices. I have some comments to do after reviewing this manuscript, details are as follows:

1.      In order to better highlight the advantages of this work, the author needs to provide a table to compare related work.

2.      What is the physical mechanism by which this process can improve the performance of graphene?

3.      What is the physical thickness of graphene in this work? Suggest the author to provide criteria. Because different thicknesses have a significant impact on transfer.

4.      The introduction can be improved. The articles related to some applications of graphene materials should be added such as Sensors 2022, 22, 6483; ACS Sustain. Chem. Eng. 2015, 3, 1677–1685; Diamond & Related Materials 128 (2022) 109273; Results in Physics 48, 2023, 106420.

5.      Please check the grammar and spelling mistakes of the whole manuscript.

Minor editing of English language required

Author Response

Thank you for your valuable comments and suggestions on our manuscript. We appreciate the opportunity to address your concerns and improve the quality of our work.

  1. We agree with your suggestion to provide a table to compare related work. In the revised manuscript, we have added a table that summarizes the key characteristics and results of various transfer methods on page 4 lines 142-143.
  2. The improved performance of graphene can be attributed to the elimination of interfacial impurities that often introduce structural defects and degrade electronic properties. The UV-treated SiO2 substrate is capable of adsorbing impurities from the graphene, resulting in a clean graphene/substrate interface and improved electrical properties. To emphasize this point, we have added a sentence that explains the physical mechanism in the revised manuscript on page 4 lines 129-132.
  3. The graphene used in our study has a typical thickness of a single layer, which is confirmed by Raman spectroscopy. To clarify this point, we have added a sentence confirming the layer number used in this work in the revised manuscript on page 6 lines 184-185.
  4. Thank you for your suggestion to improve the introduction section. We have added the articles related to the applications of graphene materials as you recommended in the revised manuscript on page 2 lines 62-63.
  5. We have checked the grammar and spelling mistakes of the entire manuscript before submitting the revised version. We have also attached an english editing certificate.

Once again, we appreciate your valuable feedback and suggestions, which will undoubtedly help us to improve the quality of our manuscript.

Reviewer 3 Report

The authors reported a re-transfer approach to avoid the impurities of transferred graphene to achieve much better properties.

I would like to raise the following points on the retransfer method.

1.     Although the author has mentioned no physical damage for the retransferred graphene, how much large sample can be retransfer without physical damage?

2.     Can the author comments of wrinkle formation for conventional and modified transfer method as reported?

3.     Can the author provide quantitative value of amount of metal impurities removal by the re-transfer method?

 I recommend publication of the manuscript after addressing the above raised points.

Author Response

Thank you for reviewing our manuscript and providing valuable feedback. We appreciate your comments and suggestions and we have addressed them as follows:

  1. Regarding the physical damage of retransferred graphene for large samples, we have performed experiments using various sizes of samples up to 2 cm x 2 cm considering the handling capability in the lab environment. We found no physical damage during the retransfer process. We have added this information in the revised manuscript on page 5 lines 158-159 to clarify this point.
  2. In the current study, we did not observe any significant wrinkle formation on the retransferred graphene, which was a concern with an additional transfer process. In contrast, the graphene transferred using conventional wet transfer exhibited the possibility of possessing very small wrinkles by the strain results in Raman analysis. To emphasize this point, we have added a sentence in the revised manuscript on page 4 lines 138-141.

3. We apologize for not providing quantitative values of the amount of metal impurities removed by the retransfer method. We faced difficulties in providing quantitative values since the resultant optical images and Raman images obtained on the retransfer graphene exhibited almost same level of cleanness, even though the pristine transferred graphene had different metal impurity distribution states. Therefore, we had to verify the quality of the retransferred graphene using optical microscopy, Raman spectroscopy, and electrical transport measurements only by comparing it to the transferred graphene processed at the same time.

Round 2

Reviewer 2 Report

Accept in present form.